# Machine Learning-Assisted Dual-Marker Detection in Serum Small Extracellular Vesicles for the Diagnosis and Prognosis Prediction of Non-Small Cell Lung Cancer

**DOI:** 10.3390/nano12050809

**Published:** 2022-02-28

**Authors:** Wenzhe Li, Ling Zhu, Kaidi Li, Siyuan Ye, Huayi Wang, Yadong Wang, Jianchao Xue, Chen Wang, Shanqing Li, Naixin Liang, Yanlian Yang

**Affiliations:** 1CAS Key Laboratory of Standardization and Measurement for Nanotechnology, CAS Key Laboratory of Biological Effects of Nanomaterials and Nanosafety, CAS Center for Excellence in Nanoscience, National Center for Nanoscience and Technology, Beijing 100190, China; liwenzhe@pku.edu.cn (W.L.); zhul@nanoctr.cn (L.Z.); yesy@nanoctr.cn (S.Y.); wanghuayi@cibr.ac.cn (H.W.); wangch@nanoctr.cn (C.W.); 2State Key Laboratory of Natural and Biomimetic Drugs, School of Pharmaceutical Sciences, Peking University, Beijing 100871, China; 3University of Chinese Academy of Sciences, 19 A Yuquan Rd, Shijingshan District, Beijing 100049, China; 4Department of Thoracic Surgery, Peking Union Medical College Hospital, Chinese Academy of Medical Sciences and Peking Union Medical College, Beijing 100730, China; lkdrs@wchscu.cn (K.L.); pumc_wangyadong@student.pumc.edu.cn (Y.W.); eraser_1993@163.com (J.X.); lishanqing@pumch.cn (S.L.); 5Department of Chemistry, Tsinghua University, Beijing 100084, China; 6Institute for Brain Research (CIBR), Beijing 102206, China

**Keywords:** small extracellular vesicle, non-small cell lung cancer, diagnosis, prognosis prediction, machine learning

## Abstract

Small extracellular vesicles (sEVs) carry molecular information from their source cells and are desired biomarkers for cancer diagnosis. We establish a machine learning-assisted dual-marker detection method to analyze the expression of epidermal growth factor receptor (EGFR) and C-X-C chemokine receptor 4 (CXCR4) in serum sEVs for the diagnosis and prognosis prediction of non-small cell lung cancer (NSCLC). We find that the serum sEV EGFR and CXCR4 are significantly higher in advanced stage NSCLC (A/NSCLC) patients compared to early stage NSCLC (E/NSCLC) patients and the healthy donors (HDs). A receiver operating characteristic curve (ROC) analysis demonstrates that the combination of EGFR and CXCR4 in serum sEVs as an efficient diagnostic index and malignant degree indicator for NSCLC. Machine learning further shows a diagnostic accuracy of 97.4% for the training cohort and 91.7% for the validation cohort based on the combinational marker. Moreover, this machine leaning-assisted serum sEV analysis successfully predicts the possibility of tumor relapse in three NSCLC patients by comparing their serum sEVs before and three days after surgery. This study provides an intelligent serum sEV-based assay for the diagnosis and prognosis prediction of NSCLC, and will benefit the precision management of NSCLC.

## 1. Introduction

Lung cancer is the leading cause of cancer-related death worldwide, and non-small cell lung cancer (NSCLC) constitutes approximately 85% of lung cancer [1,2]. Accurate non-invasive diagnosis and early prognosis prediction based on biomarker detection help with precision medicine and prolong the survival of NSCLC. Traditional investigation of biomarkers in NSCLC is mainly based on immunohistochemistry (IHC) and fluorescent in situ hybridization (FISH) analysis of the tumor tissue. However, the difficulties in collecting tissue biopsies for repeated detection and the single biopsy bias due to intratumoral heterogeneity limit the application of the tissue-based assessment for accurate diagnosis, dynamic monitoring and prognosis prediction of NSCLC [3,4]. Recently, next generation sequencing (NGS) of the tumor tissue and plasma ctDNA has been increasingly accepted as an effective method to detect NSCLC mutation and to guide therapy in the clinical practice [5,6]. However, the detection cost is pretty high. Moreover, ctDNA is easy to be degraded, thus a large quantity of sample is needed to allow sufficient copy numbers of ctDNA derived mutations [7]. Small extracellular vesicles (sEVs) are 30–150 nm cell-released vesicles that are widely present in various body fluids such as the blood, urine, saliva, etc. [8,9,10]. They carry and transfer multiple information (e.g., proteins, nucleic acids, lipids) from their source cells [11] and serve as desired liquid biopsy biomarkers for cancer detection and therefore provide opportunities for the precision treatment of NSCLC. Due to the complex mechanism under tumor initiation and progression, each biomarker has its distinct role and diagnostic significance. Single-marker analysis on the sEVs can hardly achieve high sensitivity and specificity in the diagnosis and prognosis prediction of NSCLC. Combinational marker analysis would help to improve the diagnostic and prognostic accuracy. The epidermal growth factor receptor (EGFR) is overexpressed in 40% to 80% of NSCLC and is associated with time to progression (TTP) and overall survival (OS) of NSCLC [12,13,14]. C-X-C chemokine receptor 4 (CXCR4) is a chemokine receptor that promotes tumor progression and metastasis [15]. It is overexpressed in various cancers including NSCLC, especially the advanced NSCLC (A/NSCLC) [16,17].

In this study, we established a machine learning-assisted dual-marker detection method based on microbead enrichment and signal amplification in flow cytometry to analyze the expression of EGFR and CXCR4 and in serum sEVs for the diagnosis and prognosis prediction of NSCLC. We have previously developed a microbead-based method in diagnosis and molecular phenotyping of breast cancer which overcame the problem that the nanoscale size of the sEVs exceeded the detection limit of the traditional flow cytometry, while the analysis approach is simple and traditional [18]. In this study, we mainly focused on the intelligent and automated analysis of the detection results. A machine learning algorithm was developed based on EGFR and CXCR4 expression on serum sEVs to achieve automatic classification of healthy donors (HDs) and NSCLC patients with different malignancies. The dual biomarker analysis offered a high accuracy (97.4% for training cohort and 91.7% for validation cohort) for differentiating early stage NSCLC (E/NSCLC) from A/NSCLC and HDs, and showed potential in predicting the prognosis as early as three days after surgery. These results showed the application potential of this machine learning-assisted dual sEV marker analysis for the accurate diagnosis and prognosis prediction of NSCLC. 

## 2. Materials and Methods

### 2.1. Cell Culture

All the cell lines were obtained from the Cell Resource Center, Peking Union Medical College (Beijing, China). H1975, A549, and H1650 were cultured in Gibco™ RPMI 1640 Medium (GIBCO-BRL, Gaithersburg, MD, USA) supplemented with 10% fetal bovine serum (FBS) (GIBCO-BRL, Gaithersburg, MD, USA) and 1% penicillin/streptomycin (GIBCO-BRL, Gaithersburg, MD, USA). SW620 was cultured in Dulbecco’s Modified Eagle Medium (DMEM) (GIBCO-BRL, Gaithersburg, MD, USA) supplemented with 10% FBS (GIBCO-BRL, Gaithersburg, MD, USA) and 1% penicillin/streptomycin (GIBCO-BRL, Gaithersburg, MD, USA). The cells were maintained in a humidified inhibitor at 37 °C with 5% CO_2_—95% air. Subcultivation of the cells was performed using 0.25% trypsin and 5 mM ethylenediaminetetraacetic acid (EDTA) (GIBCO-BRL, Gaithersburg, MD, USA).

### 2.2. Clinical Samples

Human peripheral blood samples collected from healthy donors and NSCLC patients and paraffin-embedded lung sections from NSCLC patients were all obtained from the department of thoracic surgery, Peking Union Medical College Hospital, China. The collection of human samples was approved by the Medical Ethical Committee of the Peking Union Medical College Hospital (JS-1263). All the participants, including 33 NSCLC patients and 18 healthy volunteers, were recruited with informed consent. The diagnostic criteria and the demographic details of the patients are described in the supplementary information (Appendix A). Serum samples and metastatic tumor specimens were collected after the lung cancer was pathologically confirmed, and before any chemo-/radio- therapies. The peripheral blood was collected in blood collection tubes and was allowed to clot for 30 min at room temperature. The serum was separated by centrifugation at 3000 rpm for 10 min, aliquoted and stored at −80 °C prior to use. Tumor specimens collected by surgical removal was embedded in 4% paraformaldehyde at room temperature for at least 24 h. After dehydration and paraffin embedding, the tumor specimens were sliced into paraffin sections using a rotary microtome (Leica RM2265, Nussloch, Germany) and IHC assessment was performed on the freshly prepared specimen sections.

All of the samples were randomly selected from larger cohorts and were analyzed in a blinded fashion. Unblinding of clinical parameters and corresponding experimental data was performed only after finishing all experiments.

### 2.3. sEV Purification

sEVs were purified by ultracentrifugation according to a previously described procedure with modifications [19,20]. To isolate sEVs from the cell culture supernatant, the cell culture medium was changed to a medium supplemented with 10% EV-free FBS (GIBCO-BRL, Gaithersburg, MD, USA) and 1% penicillin/streptomycin (GIBCO-BRL, Gaithersburg, MD, USA) for 48 h before sEV purification. To isolate sEVs from human sera, 500 μL of human sera was diluted with PBS solution to the final volume of 27 mL before sEV purification. The prepared cell culture medium or the diluted human sera was collected into 50-mL centrifuge tubes and centrifuged successively at 800× *g* for 5 min and 2000× *g* for 10 min, followed by filtration through a 0.22-μm filter (Merck Millipore, Darmstadt, Germany) to eliminate large dead cells and cell debris. The conditioned medium was then transferred to the 26.3-mL polycarbonate ultracentrifugation tubes matching the 70-Ti rotor and was ultracentrifuged at 100,000× *g* for 2 h at 4 °C to purify sEVs from the cell culture medium and at 150,000 g overnight at 4 °C to purify sEVs from the human sera (OPTIMA XPN-100, Beckman Coulter Inc, Brea, CA, USA) due to the high viscosity of sera. The supernatant was removed completely and the pellet was washed with PBS and ultracentrifuged (at 100,000× *g* to purify sEVs from the cell culture medium and at 150,000× *g* to purify sEVs from the human sera) for 2 h at 4 °C for a second time. The purified sEV pellets were resuspended in 100 μL PBS.

The size and morphology of the purified sEVs were characterized by nanoparticle tracking analysis (NTA), transmission electron microscopy (TEM) and scanning electron microscopy (SEM) as described below.

### 2.4. Nanoparticle Tracking Analysis 

The number and size distribution of sEVs were measured using the NanoSight LM14 system with a 405 nm laser (NanoSight Technology, Malvern, UK). The sEVs derived from cell cultures and sera were diluted in PBS to keep the concentration at 10^8^–10^9^ particles/mL. Samples were injected into the sample chamber with a syringe, measured in triplicate with a high-sensitivity scientific complementary metal-oxide semiconductor (sCMOS) camera at camera setting 16 with an acquisition time of 60 s and a detection threshold setting of 7. The sample chamber was rinsed three times between measuring different samples. Finally, the data were analyzed using the nanoparticle tracking analysis software (NTA version 2.3; Malvern Instruments, Malvern, UK).

### 2.5. Transmission Electron Microscopy

An optical concentration of sEVs or sEVs-bound beads were loaded onto 200-mesh carbon/formvar coated grids (Beijing Zhongjingkeyi Technology Co., Ltd., Beijing, China) and were allowed to absorb on the grids for 20 min, followed by negative staining with uranyl acetate for 10 min. After rinsing with PBS, the grids were air-dried and subsequently observed with a Hitachi transmission electron microscope.

### 2.6. Scanning Electron Microscopy

Isolated sEVs were loaded onto silicon wafers and dried in a drying oven, followed by sputter-coating with a thin layer of gold. SEM images were obtained using a Hitachi S-3400N scanning electron microscope (Hitachi High-Tech, Tokyo, Japan) at an acceleration potential of 15 kV.

### 2.7. Flow Cytometry Analysis

For flow cytometry analysis based on EV-bound beads, 4 μg sEVs were attached to 1 μL 4-μm aldehyde/sulphate latex beads (Invitrogen, Waltham, MA, USA) for 1 h at room temperature with continuous rotation (the sEV/beads ratio is determined by the saturation assay in Appendix A). The input of sEVs was normalized by total protein content on the sEVs according to relative protein quantification using a bicinchoninic acid (BCA) kit (Solarbio, Beijing, China). The reaction was stopped with 100 μM glycine and left rotating for 30 min at room temperature. EV-bound beads were washed once in 0.5% Bovine Serum Albumin (BSA)/PBS and blocked with 5% BSA/PBS with rotation at room temperature for 1 h, then washed a second time in 0.5% BSA/PBS and incubated with anti-EGFR (rabbit mAb, Cell Signaling Technology (CST), #4267) and anti- CXCR4 (goat mAb, Abcam, ab1670, Cambridge, UK) when rotating at 4 °C for 1 h. Beads were centrifuged for 3 min at 14,800× *g*, the supernatant was discarded and beads were washed in 0.5% BSA/PBS, then incubated with Alexa-647 (Abcam, anti-rabbit, ab150107) or Alexa-488 (Abcam, anti-goat, ab150073) tagged secondary antibodies with 30 min rotation at 4 °C. After blocking with 5% BSA/PBS, secondary antibodies were incubated with the EV-bound beads as controls as described in the previous studies [19,21,22]. The samples were finally washed by 0.5% BSA/PBS three times and re-suspended in 200 μL PBS. Flow cytometry analysis was performed on a BD Accuri^TM^ C6 Flow Cytometer (BD Bioscience, Franklin Lakes, NJ, USA).

### 2.8. Protein Separation and Western Blot Analysis

Cells or sEVs were lysed with lysis buffer, supplemented with protease inhibitor cocktail and phenylmenthysulfonyl fluoride (Thermo Scientific, Waltham, MA, USA) on ice for 60 min. Protein fractions were collected by centrifugation and were normalized according to relative protein quantification using a BCA protein assay kit (Solarbio). Proteins were separated in NuPAGE 10% Bis-Tris Gels (Thermo Scientific, Waltham, MA, USA) under reducing condition, and transferred onto polyvinylidene difluoride (PVDF) membranes (0.45 μm, Millipore, Bedford, MA, USA). The membranes were blocked with 5% non-fat milk (BD Bioscience, Franklin Lakes, NJ, USA) in Tris-buffered saline with 0.1% Tween (TBST) for 1 h and then incubated with anti-EGFR (rabbit mAb, CST, #4267, Boston, MA, USA), anti-CXCR4 (goat mAb, Abcam, ab1670, Cambridge, UK), anti-CD44 (rabbit mAb, Abcam, ab51037, Cambridge, UK), anti-ALDH1A1 (rabbit mAb, CST, #36671, Boston, MA, USA), anti-β actin (mouse mAb, EASYBIO, BE0037, Beijing, China), anti-CD63 (mouse mAb, Santa Cruz, sc-5275, Dallas, TX, USA), anti-CD81 (mouse mAb, Santa Cruz, sc-166029, Dallas, TX, USA), anti-flotillin (rabbit mAb, Abcam, 133497, Cambridge, UK), anti-APOA1(rabbit pAb, Sino biological, 10686-T52, Beijing, China) or anti-calnexin (rabbit pAb, Sino biological, 102109-T32, Beijing, China) overnight at 4 °C. Horse-radish-peroxidase (HRP) conjugated goat anti-rabbit IgG (CST, #7074, Boston, MA, USA), goat anti-mouse IgG (CST #7076, Boston, MA, USA) or donkey anti-goat IgG (Santa Cruz, sc-2020, Dallas, TX, USA) were used as secondary antibodies. The blots were visualized by Image Lab (BIO-RAD, Hercules, CA, USA) with an Enhanced Chemiluminescence Kit (Thermo Pierce, Waltham, MA, USA). 

### 2.9. Immunohistochemical Analysis

Tissues were sectioned into 5 μm thick slices using a microtome and transferred into adhesive slides, dried, deparaffinized in xylene and rehydrated in graded alcohol. Antigen retrieval was performed in a citrate buffer (pH 6) for 15 min after. After blocking with 5% normal goat serum (Solarbio), following staining using EGFR antibody (rabbit mAb, CST, #4267), or CXCR4 antibody (goat mAb, Abcam, ab51037). IHC staining was done using a Vectastain Elite avidin-biotin complex detection kit (Vector Laboratories), and sections were developed by DAB (Sigma-Aldrich, Darmstadt, Germany) according to the manufacturer’s recommendations. Sections were rinsed in tap water, counterstained, cleared and mounted. The image screening and photography of sections were performed using a EVOS^®^ XL Core Imaging System (Thermo Fisher Scientific, Waltham, MA, USA).

### 2.10. RNA Extraction and Real-Time Polymerase Chain Reaction (PCR)

Total RNA was extracted from sEVs using TRIzol (Life Technologies, Waltham, MA, USA) according to the manufacturer’s instructions. The first-strand cDNA was synthetized by RNA reverse-transcription using QuantScript RT Kit (TIANGEN) before Quantitative Realtime PCR (qPCR) was performed on a Realtime PCR System (Eppendorf), using SuperReal PreMix Plus (SYBR Green) (TIANGEN) according to the manufacturer’s directions. All of the reactions were run in triplicate. The mRNA levels were normalized to glyceraldehyde 3-phosphate dehydrogenase (GAPDH). The relative mRNA expression normalized to control was calculated with the equation 2^(−∆Ct), in which ∆Ct = Ct − Ct(control).

### 2.11. Logistic Regression

The logistic regression algorithm is a generalized linear model which was used in this work to compute a weighted sum of the expression of EGFR and CXCR4 of sera sEVs. We used the binary logistic regression in SPSS (Statistical Product and Service Solutions) statistical software to weigh the combination of the two biomarkers. Receiver operating characteristic (ROC) analysis was used to evaluate the specificity and sensitivity of EGFR, CXCR4 and the biomarker combination in distinguishing HDs from NSCLC patients, E/NSCLC from A/NSCLC and HDs from E/NSCLC. The area under the ROC curve (AUC) was estimated for each biomarker. All ROC analyses were performed using SPSS statistical software, and the cut-off value was determined using the Youden index.

### 2.12. Machine Learning

To choose an appropriate classification algorithm for the combinational biomarker of sera sEVs, the cross validation was performed using the whole 51 sEVs samples from HDs and NSCLCs. By comparing the classification performance with different algorithms, Random Forest, which is one of the most powerful machine learning algorithms, was finally chosen as the classification model. Comparing with some “single algorithm” such as SVM (Support Vector Machine) and Decision Tree, Random Forest is an ensemble learning algorithm containing many decision trees, which means better classification and prediction efficacy. The program was written in the Scikit-Learn library in the Python language. In the program, 51 sera samples, including 18 HDs, 16 E/NSCLC and 17 A/NSCLC patients were randomly divided into two groups. One group is a training cohort including 39 samples (14 HDs, 12 E/NSCLC and 13 A/NSCLC patients) and the other group is a validation cohort including 12 samples (4 HDs, 4 E/NSCLC and 4 A/NSCLC patients). EGFR and CXCR4 expression are imported as two independent variables, and the HDs, E/NSCLC and A/NSCLC patients were divided into three classes, termed as 0, 1, and 2, respectively. After learning using the training set database, the efficacy of the classification algorithm was validated by the validation cohort. The performance of classification in both the training and validation sets was evaluated by the accuracy. 

### 2.13. Statistical Analysis

The GraphPad Prism version 6.0 (GraphPad Software) was used for the analysis of flow cytometry results, and data were presented as the mean ± SD in the scatter plots. Comparisons between two groups were made using a Student’s *t*-test. An ROC analysis was used to evaluate the specificity and sensitivity of EGFR, CXCR4 and combinational marker of sEVs in differentiating E/NSCLC, A/NSCLC and HDs. The area under the ROC curve (AUC) was estimated for each biomarker. All ROC analyses were performed using SPSS statistical software. The cut-off value was determined using the Youden index. We have outlined the methods of our experiments on EV-TRACK (evtrack.org). The resulting link is http://evtrack.org/review.php (accessed on 18 January 2022), and the EV-TRACK ID is EV190065.

## 3. Results

### 3.1. Signal-Amplified Detection of Protein Expression on the sEVs

Protein expression on the sEVs was investigated by flow cytometry. As the nano-scaled size of sEVs exceeds the detecting limitation of traditional flow cytometry, we utilized microbead enrichment followed by dual staining for signal amplification. The sEVs were enriched on aldehyde/sulphate latex microbeads (diameter 4 μm). The enriched EGFR+ or CXCR4+ sEVs were further labeled with anti-EGFR or anti-CXCR4 and the fluorescent-tagged secondary antibody, leaving EGFR- and CXCR4- sEVs on the beads unlabeled (Figure 1A). In this way, the microbead-EV complexes could yield detectable signals for flow cytometry analysis of the protein expressions on the sEVs. We isolated the sEVs from the cell culture supernatant by differential centrifugation [19]. Three NSCLC cell lines, A549, H1650, and H1975, and one colorectal cancer cell line, SW620 (with different EGFR and CXCR4 expression) [23,24] were chosen for comparison. TEM images revealed the typical vesicle structures with a diameter of 50–150 nm (Figure 1B) corresponding to the morphology of sEVs as previously described [25]. NTA showed that the size of these particles was 130.5 ± 42.5 nm (Figure 1C, Appendix A), corresponding to the size of sEVs as previously reported [26]. 

TEM and SEM characterizations showed that the microbeads were coated with sEVs (Figure 1D,E, Appendix A), indicating the enrichment of sEVs on the microbeads. We further performed a saturation assay to ensure that the microbeads were saturated with the captured sEVs. Different quantities of sEVs from A549 cells and SW620 cells were captured on the microbeads, and were labeled with FM^TM^4-64FX, a dye of the sEV membrane lipids, for the subsequent fluorescence intensity quantification by flow cytometry. The sEVs from both cell lines showed a saturation ratio of ~2 μg sEVs/μL beads (Appendix A), far below the EV/bead ratio we used in our flow cytometry analysis of sEVs (4 μg sEVs/μL beads), indicating the saturation of sEVs on the beads.

### 3.2. Expression of EGFR and CXCR4 in sEVs Represent the Ones in the Cells of Origin

We checked if the expression of EGFR and CXCR4 on the sEVs represent the ones in the source cells. A flow cytometry analysis showed that the expression of EGFR was the highest in A549, moderate in H1975, and the lowest in SW620 (Figure 2A, Appendix A), in accordance with the reported expression level of EGFR in these cell lines [23,24]. The expression of EGFR in the cell-derived sEVs correlated well the ones in the source cell (Figure 2A, Appendix A). Similarly, the expression of CXCR4 in sEVs was the highest in H1975, moderate in A549, and the lowest in H1650, correlating with the expression level of CXCR4 in the cell lines (Figure 2B, Appendix A). These results were confirmed by western blot analysis showing that the expression of EGFR and CXCR4 in the cell-derived sEVs correlated well with the one in the source cells (Figure 2C,D, Appendix A). As guided by the minimal experimental requirements for the definition and functional studies of sEVs provided by the International Society for Extracellular Vesicles (MISEV) [27], two EV-enriched marker proteins, CD81, and flotillin-1, were used as the positive control, and the cell-derived calnexin was used as the negative control to show the purity of the isolated sEVs. To confirm that the flow cytometry signals were not from the non-specific binding of antibodies to the beads, we incubated anti-EGFR or anti-CXCR4 and the fluorescent-tagged secondary antibody directly to the blocked beads without sEV binding. We observed no signal in the flow cytometry analysis (Appendix A), demonstrating that the signals in the flow cytometry analysis were from the sEVs bound to the beads, rather than the non-specific binding of antibodies to the beads. Immunogold TEM of sEVs derived from A549 cells using antibodies specific to EGFR and CXCR4 also showed the binding of immunogold nanoparticles on the surface of sEVs (Appendix A), demonstrating the positive expression of EGFR and CXCR4 in the sEVs from A549 cells. These results, taken together, indicated that EGFR and CXCR4 in sEVs could well represent the expression of these proteins in the cells of origin, enabling them to be candidate biomarkers for NSCLC.

### 3.3. EGFR and CXCR4 in Serum sEVs as Biomarkers for NSCLC Diagnosis and Staging

We then checked if EGFR and CXCR4 in sEVs derived from serum could act as tumor detection biomarkers for NSCLC. sEVs were isolated from the sera of 51 histologically validated NSCLC adenocarcinoma patients, including 16 patients at stage I (termed early stage, E/NSCLC), 17 at stage II-IV (termed advanced stage, A/NSCLC), and 18 healthy donors (HDs) using differential centrifugation [20]. TEM and NTA analysis revealed that the morphology and size of the isolated vesicles were characteristic of the ones of sEVs, demonstrating that sEVs were successfully isolated from human sera and were captured by the aldehyde/latex beads (Appendix A).

The expression of EGFR and CXCR4 in sera sEVs were analyzed by microbead-assisted flow cytometry as described above (Figure 3A,B). The average expression levels of EGFR and CXCR4 in sEVs were both significantly higher in A/NSCLCs compared to E/NSCLCs and HDs, indicating the diagnostic significance of sEV EGFR and CXCR4 as biomarkers to identify A/NSCLCs. We compared the expression of EGFR and CXCR4 in serum sEV with that in the primary tumor tissue assessed by IHC, which is a clinically used gold standard to examine the expression of marker proteins. Four NSCLC patients with different expression levels of EGFR and CXCR4 on serum sEVs were selected for comparison. We found that the expression levels of EGFR and CXCR4 in serum sEVs correlated well with the IHC staining results of the primary tumor tissue for the four patients. The E/NSCLC patients, one with 3.4% EGFR+ and 10.5% CXCR4+ serum sEVs and the other with 4.1% EGFR+ and 12.6% CXCR4+ serum sEVs, both had nearly negative EGFR staining and low CXCR4 staining in the primary tumor tissue (Figure 3C and Appendix A). The A/NSCLC patients, one with 38.4% EGFR+ and 45.3% CXCR4+ serum sEVs and the other with 23% EGFR+ and 61.6% CXCR4+ serum sEVs both showed high levels of EGFR and CXCR4 in the primary tumor biopsy (Figure 3D and Appendix A). These results suggested that the expression levels of EGFR and CXCR4 in serum sEVs may represent the expression levels of those in the primary tumor tissues, showing the potential of this easy flow cytometry-based protein profiling method in clinical application. 

### 3.4. Bivariate Analysis and Machine Learning of EGFR and CXCR4 Expression on Sera sEVs Showing High Accuracy for the Diagnosis and Prognosis Prediction of NSCLC

To evaluate the discriminative efficacy of EGFR and CXCR4 in serum sEVs to diagnose NSCLC, we performed ROC analysis on serum sEV EGFR and CXCR4 in classifying NSCLC patients (including E/NSCLC, n= 16 and E/NSCLC, n = 17 and HDs, n = 18). We found that EGFR in serum sEVs can achieve high sensitivity and specificity in differentiating A/NSCLCs from E/NSCLCs (area under the curve (AUC) 0.960, 95% confidential interval (95% CI) 89.5–100%, sensitivity 94.1%, specificity 93.8%) and A/NSCLCs from HDs (AUC 0.977, 95% CI 93.8–100%, sensitivity 94.1%, specificity 94.4%), and moderate sensitivity and specificity in differentiating NSCLCs from HDs (AUC 0.778, 95% CI 65.3–90.3%, sensitivity 60.6%, specificity 88.9%) (Figure 4A–C, Appendix A). CXCR4 in serum sEVs had moderate discriminative efficacy in identifying A/NSCLCs and E/NSCLCs (AUC 0.842, 95% CI 70.9–97.5%, sensitivity 76.5%, specificity 81.3%), and in identifying A/NSCLCs patients and HDs (AUC 0.815, 95% CI 67.7–95.4%, sensitivity 82.4%, specificity 72.2%), but insufficient sensitivity and specificity in classifying NSCLCs and HDs (AUC 0.668, 95% CI 51.4–82.2%, sensitivity 60.6%, specificity 72.2%) (Figure 4A–C, Appendix A). 

Considering the different roles of the two biomarkers in tumor initiating and progression, we performed bivariate analysis to check if the combination of EGFR and CXCR4 can achieve improved diagnostic capability. Using a logistic regression algorithm, a weighted sum of the expression of EGFR and CXCR4 on sera sEVs was calculated and was defined as the combinational marker. The diagnostic effectiveness of the combinational marker was also evaluated by ROC analysis, and the results showed that the AUC is 0.963 (95% CI 90.4–100%, sensitivity 94.1%, specificity 93.8%) for differentiating E/NSCLC from A/NSCLC, 0.983 (95% CI 95.2–100%, sensitivity 94.1%, specificity 94.4%) for differentiating E/NSCLC from HDs, and 0.785 (95% CI 66.0–90.9%, sensitivity 48.5%, specificity 100%) for differentiating NSCLC patients from HDs, all of which were better than those of the single-marker analyses (Figure 4A–C, Appendix A). It is worth noting that the specificity to distinguish NSCLCs from HDs was elevated from 88.9% using single EGFR as the classification index to 100% using the combinational marker as the classification index. These findings confirmed our assumption that the combinational marker is more sufficient than either of the single markers in classifying HDs, E/NSCLC and A/NSCLC patients, and particularly efficient for diagnosing the A/NSCLC patients and staging E/NSCLC and A/NSCLC patients.

To establish an intelligent and automated method for accurate differentiation of HDs, E/NSCLC, and A/NSCLC, EGFR and CXCR4 expression in sEVs were used as machine learning objects, and sera sEVs from 51 individuals were randomly divided into the training cohort (39 samples) and validation cohort (12 samples). We defined the group of HDs, E/NSCLC, and A/NSCLC as 0, 1, 2, respectively, in the machine learning algorithm and input EGFR and CXCR4 expression of every serum sEV sample. After repeatedly optimizing parameters, the two protein biomarkers finally achieved an accuracy of 97.4% for discrimination of HDs (n = 14), E/NSCLC (n = 12) and A/NSLCLC (n = 13) patients in the training cohort (Figure 4D). The validation cohort with a limited number of 12 samples showed an accuracy of 91.7% for classification of HDs (n = 4), E/NSCLC (n = 4), and A/NSCLC (n = 4) (Figure 4E). 

We further explored the capacity of sEVs in early prediction of the recurrence risk of NSCLC patients after surgery by applying the algorithm to EGFR and CXCR4 of serum sEVs isolated from three follow-up patients. Serum sEVs were isolated from the three NSCLC patients before and three days after surgery (#1, #2, #3), and EGFR and CXCR4 expression on the sEV were analyzed (Figure 4F, upper). By inputting the EGFR and CXCR4 expression data to the machine learning algorithm, classification results for the sEVs of 3 NSCLC patients before and three days after surgery were obtained which were in complete agreement with the clinical IHC results before and six months after surgery (Figure 4F, lower table). Specifically, multiple small pulmonary nodules were observed six months after surgery for patient #3 whose serum sEVs isolated three days after surgery was classified as class 1, which means patient class. Machine learning outcomes of sera sEVs from patients #1 and #2 three days after surgery were judged as class 0, which was in accordance with the favorable prognosis of the two patients with no sign of recurrence. These results suggested that the combination of EGFR and CXCR4 may be an effective method to monitor and predict prognostics of NSCLC patients after surgery, and it may provide information to suggest tumor recurrence much earlier than the routine clinical evaluation. With the aid of machine learning prediction outcomes, doctors can develop reasonable and personalized therapeutic strategies to avoid tumor relapse. Collectively, these results suggested that the combination of EGFR and CXCR4 expression in serum sEVs may serve as an independent marker for NSCLC diagnosis, monitoring, and prognosis prediction. Furthermore, machine learning analysis using the combinational markers of these two proteins could greatly improve the diagnosis effectiveness and also automate the analysis process.

## 4. Discussion

sEVs have attracted increasing attention with regard to liquid biopsies that are part of the cancer diagnosis. However, given the complex tumor initiation and progression mechanisms, single-marker analysis on sEVs can hardly achieve high diagnostic and prognostic accuracy. Here we established a dual-marker detection method to analyze the expression of EGFR and CXCR4 on serum sEVs for the diagnosis and prognosis prediction of NSCLS. sEVs were enriched on microbeads and stained with fluorescent antibodies against EGFR and CXCR4 to facilitate signal amplification of these two proteins on the sEVs in flow cytometry analysis, overcoming the problem whereby the nano-scaled size of the sEVs exceeds the detection limit of the traditional flow cytometry. We demonstrated that the expression levels of EGFR and CXCR4 on the sEVs well represented the ones in the source lung cancer cells. We compared serum sEVs from the HDs, E/NSCLCs and A/NSCLCs, and found that the expressions of EGFR and CXCR4 on serum sEVs were significantly higher in A/NSCLCs compared to HDs and E/NSCLCs, suggesting the capability of serum sEV EGFR and CXCR4 for the diagnosis of NSCLC. Moreover, the expression level of EGFR and CXCR4 in serum sEVs correlated well with the ones in the primary tumor tissues as assessed by IHC, suggesting that sEV-based assessments could be used as a noninvasive surrogate to the tissue-based examination by IHC and therefore had potential in clinical application. Considering the various significance of EGFR and CXCR4 for NSCLC progression, logistic regression analysis was used to obtain an unweighted sum of EGFR and CXCR4, which was used as the combinational marker. ROC analysis revealed that the combinational marker had better performance than the single marker in discriminating NSCLC patients from HDs, especially in discriminating A/NSCLC patients from HDs, demonstrating the potential of the combined protein marker in acting as an independent detection biomarker for NSCLC. We further established an intelligent and automated sEV-based method for the accurate detection of NSCLC based on a machine learning algorithm. We found that EGFR and CXCR4 expression identified by machine learning showed an accuracy of 97.4% for the training cohort and 91.7% for the validation cohort in diagnosing and staging NSCLC patients. Moreover, utilizing this machine learning algorithm, we have successfully predicted the possibility of tumor relapse in three patients by classifying their serum sEVs before and three days after surgery. Machine learning-based prognostic classification correlated well with the real clinical prognosis, indicating the capability of the machine learning-assisted serum sEV dual-marker detection method for the early prediction of tumor recurrence after surgery.

In conclusion, the current study demonstrated that the combination of EGFR and CXCR4 in serum sEVs can act as efficient liquid biopsy biomarkers and that machine learning applied in EGFR and CXCR4 expression of serum sEVs improved the diagnostic effectiveness. This study could shed light on clinical applications of this detection method with machine learning analysis for NSCLC diagnosis and early prediction of relapse after surgery. Because of the high accuracy and intelligent characteristics, the detection platform shows clinical potential in monitoring the development of NSCLC, evaluating the prognosis, predicting the possibility of tumor recurrence and facilitating precision therapy.

## Figures and Tables

**Figure 1 nanomaterials-12-00809-f001:**
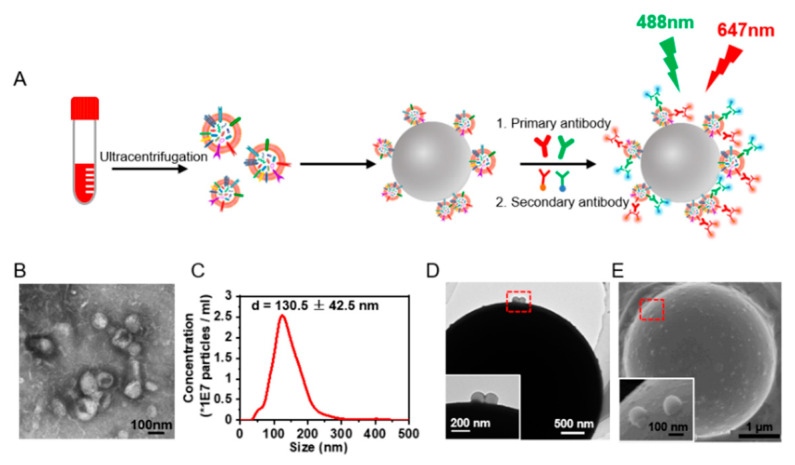
Extraction and microbead enrichment of sEVs. (**A**) Schematic illustration of microbead enrichment of sEVs from serum and immunostaining of the sEVs. The sEVs were enriched on 4-μm aldehyde/sulphate latex beads and were stained with anti-EGFR and anti-CXCR4 and the fluorescent-tagged secondary antibody to yield detectable signals for flow cytometry analysis of EGFR+ and CXCR4+ sEVs. (**B**) TEM images of sEVs released from A549 cells. (**C**) Size distribution of sEVs released from A549 cells analyzed by NTA. (**D**,**E**) TEM (**D**) and SEM (**E**) images of sEVs enriched on the microbeads. Inset: zoom-in images of the bead-bound sEVs in the red dashed box.

**Figure 2 nanomaterials-12-00809-f002:**
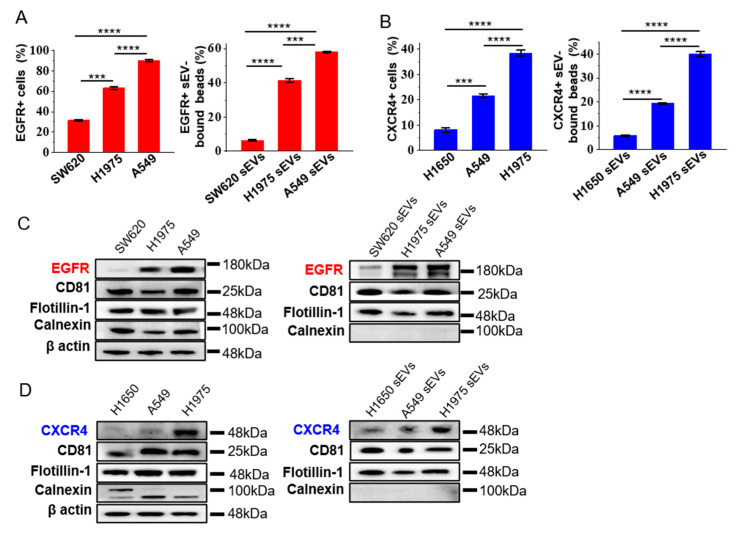
Expression levels of EGFR and CXCR4 on sEVs reflect those of the source cells. (**A**,**B**) The expression of EGFR (**A**) and CXCR4 (**B**) in tumor cell lines and cell-derived sEVs analyzed by flow cytometry. Colon cancer cell line SW620 and NSCLC cell lines H1650, H1975 and A549 were analyzed. Data are presented as mean ± S.D. *** *p* < 0.001, **** *p* < 0.0001, (Student’s *t*-test). (**C**,**D**) Western blotting of the expression of EGFR and CXCR4 in tumor cells and cell-derived sEVs. β actin was used as the loading control for cell lines. CD81 and flotillin-1 were used as positive controls, while calnexin was used as negative control for sEVs. The blots were cropped from their original images and the full-length blots are presented in Appendix A.

**Figure 3 nanomaterials-12-00809-f003:**
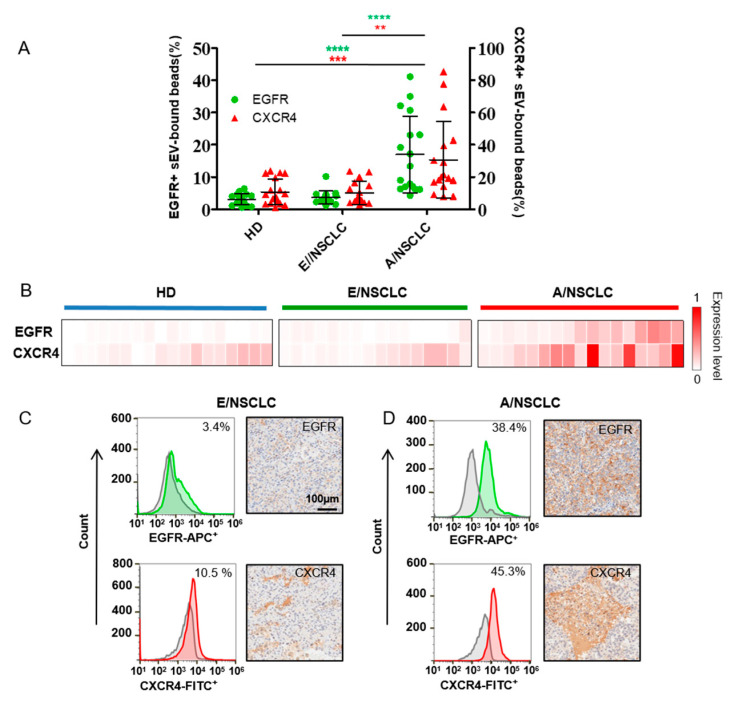
EGFR and CXCR4 expression in serum sEVs act as diagnosis and staging biomarkers of NSCLC. (**A**) Percentage of EGFR+ sEV-bound beads from healthy donors (HDs) (n = 18), early stage NSCLC patients (E/NSCLC) (n = 16) and advanced stage NSCLC patients (A/NSCLC) (n = 17) analyzed by flow cytometry. Data are presented as mean ± S.D. ** *p* < 0.01, *** *p* < 0.001, **** *p* < 0.0001, (Student’s *t*-test). (**B**) Heatmap of EGFR and CXCR4 expression profiles in sEVs from 18 HDs, 16 E/NSCLC and 17 A/NSCLC patients. Each column represents an individual sample. (**C**,**D**) The expression of EGFR or CXCR4 in serum sEVs examined by flow cytometry (left) was consistent with that in the patient-matched primary tumor tissue assessed by IHC staining (right) in one E/NSCLC patient (**C**) and one A/NSCLC patient (**D**).

**Figure 4 nanomaterials-12-00809-f004:**
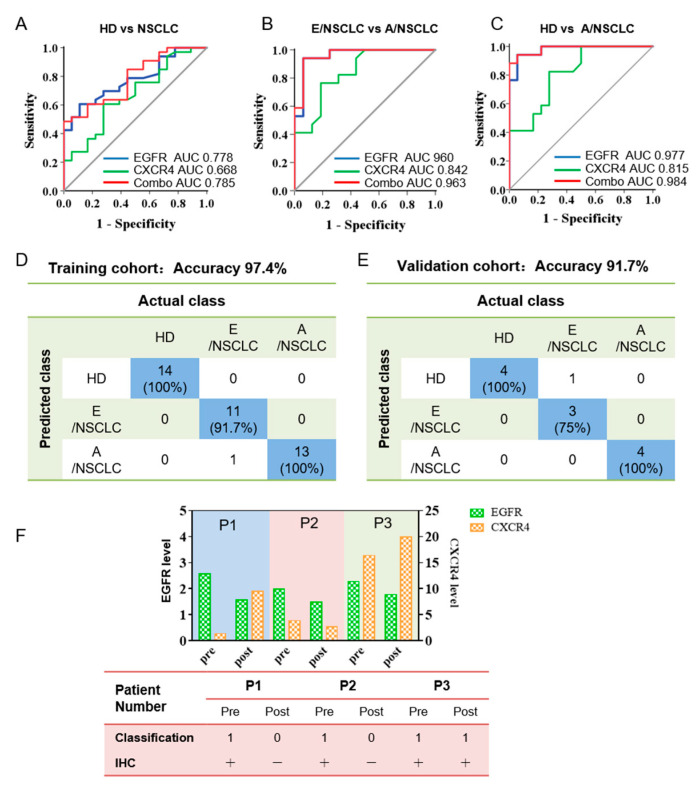
Combined analysis of EGFR and CXCR4 in serum sEVs for the classification of NSCLC by machine learning. (**A**–**C**) ROC curves showing the discriminative efficacy of EGFR, CXCR4, and the combination of the two markers in differentiating NSCLC patients from HDs (**A**), E/NSCLC from A/NSCLC patients (**B**), and A/NSCLC patients from HDs (**C**). The area under the curve (AUC). (**D**,**E**) The combination of EGFR and CXCR4 has an accuracy of 97.4% in classifying of HDs, E/NSCLC, and A/NSCLC patients for the training cohort (**D**) and 91.7% for the validation cohort (**E**) by machine learning. The data in blue boxes are the number and percentage of the truly predicted results. (**F**) Prognostic significance of serum sEVs for NSCLC progression. Upper: histogram showing the expression of EGFR and CXCR4 in serum sEVs from three stage I NSCLC patients before and three days after surgery. Lower: machine learning-based predictive classification and IHC-based validation of the prognosis of the NSCLC patients. Machine learning classification was according to the expression of EGFR and CXCR4 on serum sEVs before and three days after surgery. IHC validation was performed six months after surgery. In patient #3 who was classified as class 1 (patient class) three days after surgery by machine learning, multiple small pulmonary nodules were observed six months after surgery, indicating a prediction accuracy of 100%.

## Data Availability

We have outlined the methods of our experiments on EV-TRACK (evtrack.org). The resulting link is http://evtrack.org/review.php (accessed on 18 January 2022), and the EV-TRACK ID is EV190065.

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
