# Peer review of "Machine Learning-Assisted Dual-Marker Detection in Serum Small Extracellular Vesicles for the Diagnosis and Prognosis Prediction of Non-Small Cell Lung Cancer"

_nanomaterials, 2022, doi:10.3390/nano12050809_

Round 1

Reviewer 1 Report

The paper, entitled "Machine learning-assisted dual-marker detection in serum small extracellular vesicles for the diagnosis and prognosis prediction of non-small cell lung cancer” ” aims to establish a machine learning-assisted dual-maker detection method for EGFR and CXCR4 diagnosis and prognosis prediction in serum using. Serum sEV EGFR and CXCR4 were significantly higher in advanced stage NSCLC (A/NSCLC) patients compared to early stage NSCLC (E/NSCLC) that may help the clinical diagnosis. This work provides novel data. It has merit of publication, though a higher number of patients should provider a higher rate in validation phase.

Author Response

Response to Reviewer1 Comment

Point: The paper, entitled "Machine learning-assisted dual-marker detection in serum small extracellular vesicles for the diagnosis and prognosis prediction of non-small cell lung cancer” aims to establish a machine learning-assisted dual-maker detection method for EGFR and CXCR4 diagnosis and prognosis prediction in serum using. Serum sEV EGFR and CXCR4 were significantly higher in advanced stage NSCLC (A/NSCLC) patients compared to early stage NSCLC (E/NSCLC) that may help the clinical diagnosis. This work provides novel data. It has merit of publication, though a higher number of patients should provider a higher rate in validation phase.

Response: We agree with the reviewer that a higher number of patients should provide a higher rate in validation phase. Since the current study aimed at developing an efficient liquid biopsy method for the accurate detection of NSCLC based on machine learning algorithm, a large proportion of samples of patients were used for training and optimizing machine learning accuracy. Because of limited number of patients who meet the criteria were enrolled in limited time, the patient number of validation cohort was relatively less.

As suggested by the reviewer, we will enlarge the number of patients for validation cohort in the following studies, and several samples of patient who meet the criteria have already been collected recently.

Reviewer 2 Report

Introduction:

The authors talk about biomarkers in NSCLC without talking about NGS, which is the main technology in both, tumor samples and blood samples. The last paragraph of the introduction should be shorter and don’t talk too much about their own results that will be explained further during the manuscript.

Methods are well explained and completed. However, in the Machine Learning section, the training group it’s said to be 40 samples (14 HD, 12 E/NSCLC, and 13 A/NSCLC) (line 234-235) and the sum is 39, no 40.  I guess there is a mistake because the validation cohort is 12 (40 + 12 is 52), and here there are 51 samples.

Results:

The results are well explained. It may be a little excessive to affirm the demonstrations that levels of EGFR and CXCR4 correlated well in blood and in tumor biopsy when they only tested 2 patients. (line 341 to 351), because there is a difference in EGFR mean in sEV between Advanced NSCLC and Early NSCLS, and it could be explained by different EGFR expression in tumor biopsies (low n sample).

Discussion:

The authors should be careful when they affirm that EGFR and CXCR4 expression in sEVs could be a good biomarker for diagnostic, because the AUC of E/NSCLC vs HD is not expressed, and it may be lower than 0.5. So, it is a good biomarker for advanced NSCLC diagnostic, not for a NSCLC diagnosis.

The Machine Learning approach is nice, however, there is only 51 samples, and the validation cohort is very small (12). The percentage 80-20 for test-training is ok, but overall, it is a small cohort.

The sample is enriched of EGFR mutated (high % of females, and I guess low % of smokers), so it is easier to find differences between Adv NSCLC and HD. However, it could not be applied to a smoker lung cancer group or overall population. Another weakness could be the different ages of the HD group, Early group, or advanced group.

Last point, a liquid biomarker of sEVs could be really interesting if analysing them in clinics is easier than checking circulating tumor DNA or circulating tumor cells. Even, more biomarkers could be analyzed during the follow-up of a patient depending on what markers are expressed in the tumor. However, in this study, there is only a follow-up of 3 patients, and is not enough to reach any conclusion.

Supplementary figures S4 and S5, the WB has an extra in EGFR and CXCR4, is it a control?? There is no name of what it is.

To sum, I find it is an interesting article and well presented. However, I recommend to improve the introduction and the discussion before publishing it.

Reviewer 3 Report

In this paper, the authors report on the dual-marker detection in serum small extracellular vesicles for the diagnosis and prognosis prediction of non-small cell lung cancer using microbeads.  Overall, the innovation is a concern.  The microbead-based method has been well studied previously ("Noninvasive Diagnosis and Molecular Phenotyping of Breast Cancer through Microbead-Assisted Flow Cytometry Detection of Tumor-Derived Extracellular Vesicles", Small Methods, 2018), and the machine learning-assisted detection has been published before.  Authors did not give a through introduction of current state of art, neither the novelty in this manuscript.  The reviewer could not recommend publication in Nanomaterials.

Author Response

Response to Reviewer 3 Comments

Point: In this paper, the authors report on the dual-marker detection in serum small extracellular vesicles for the diagnosis and prognosis prediction of non-small cell lung cancer using microbeads.  Overall, the innovation is a concern.  The microbead-based method has been well studied previously ("Noninvasive Diagnosis and Molecular Phenotyping of Breast Cancer through Microbead-Assisted Flow Cytometry Detection of Tumor-Derived Extracellular Vesicles", Small Methods, 2018), and the machine learning-assisted detection has been published before.  Authors did not give a through introduction of current state of art, neither the novelty in this manuscript.  The reviewer could not recommend publication in Nanomaterials.

Response:

In the current study, we mainly focus on the machine learning-based automated analysis of NSCLC using the combinational marker calculated by logistic regression analysis, and the evaluation of the combinational marker scoring system in real clinical trials. Microbead-based technology is only a method to detect the expression of EGFR and CXCR4 in serum sEVs. The focus and significance of this manuscript are totally different from our previous study about the breast tumor detection ("Noninvasive Diagnosis and Molecular Phenotyping of Breast Cancer through Microbead-Assisted Flow Cytometry Detection of Tumor-Derived Extracellular Vesicles", Small Methods, 2018) which focuses on developing microbead-based sEV enrichment and flow cytometry-based sEV detection method. In this study, we offered an approach to get a weighted combinational marker by logistic regression analysis of multiple parameters and further established an intelligent and automated sEV-based method for the accurate detection of NSCLC by machine learning algorithm. We found that EGFR and CXCR4 expression identified by machine learning showed an accuracy of 97.4% for the training cohort and 91.7% for the validation cohort in diagnosing and staging NSCLC patients. Moreover, utilizing this machine learning algorithm, we have successfully predicted the possibility of tumor relapse in three patients by classifying their serum sEVs before and three days after surgery. In conclusion, the current study demonstrated that the combination of EGFR and CXCR4 in serum sEVs can act as efficient liquid biopsy biomarkers and a machine learning applied in EGFR and CXCR4 expression of serum sEVs provides a clinical application for NSCLC diagnosis and relapse early prediction after surgery. Since the high accuracy and intelligent characteristics, the detection platform shows clinical potential in monitoring the development of NSCLC, evaluating the prognosis, predicting the possibility of tumor recurrence and guiding precision therapy. Based on these, we believe that our study is of sufficient quality and novelty.

In the revised manuscript, we have cited and introduced the previously studied microbead-based method in the section of “Introduction” as follows: “In our previous study, we have developed a micaobead-based method in Diagnosis and Molecular Phenotyping of Breast Cancer which overcame the problem that the nanoscale size of the sEVs exceeded the detection limit of the traditional flow cytometry, while the analysis approach is simple and traditional[18]. In this study, we mainly focused on the intelligent and automated analysis of the detection results.

Round 2

Reviewer 3 Report

The authors have revised their manuscript, sufficiently.